# The Relationship between Cognitive Impairment and Social Vulnerability among the Elderly: Evidence from an Unconditional Quantile Regression Analysis in China

**DOI:** 10.3390/ijerph16193684

**Published:** 2019-09-30

**Authors:** Junkai Zhao, Xinxin Zhang, Zongmin Li

**Affiliations:** Business School, Sichuan University, Chengdu 610064, China; zhaojunkai@stu.scu.edu.cn (J.Z.); 2016141081023@stu.scu.edu.cn (X.Z.)

**Keywords:** social vulnerability, cognitive impairment, older people, unconditional quantile regression

## Abstract

As the global proportion of the elderly population has been growing rapidly, it has become important to better understand the holistic social factors involved in cognitive impairment in the elderly. To investigate the relationship between social vulnerability and cognitive impairment in the elderly, this study applied an unconditional quantile regression model on open source health survey data in China. It was used to estimate the relationship for full sample and subsamples divided by different levels of a specific covariate. It was found that the cognitive impairment had a positive association with social vulnerability, and this relationship is stronger at the higher cognitive impairment quantiles. The cognitive impairment of females and elderly who took less exercise; had lower self-rated health; had greater incidences of depression, chronic diseases, and physical limitations; and consumed less fruit and vegetables, milk and tea were more related to social vulnerability. These results provide some insights into the strategies that could be used by the elderly to decrease the risk of cognitive impairment.

## 1. Introduction

The ageing of populations has been rapidly accelerating worldwide [1]. By 2020, the elderly are estimated to exceed 1 billion, 70% of whom will be in the developing countries [1]. Population ageing in China has been and continues to be significantly faster than in many other developing countries. By 2050, it is estimated that there will be 90.4 million people aged 80 years or over in China, which will be the world’s largest elderly population in that age group [2]. As people age, their risk of health disorders grows [3,4]. Cognitive impairment in particular has been found to be an age-related condition; however, as it is influenced by many social factors, there are opportunities for positive public-health interventions [3]. To improve the health of the elderly, many countries have been providing better social conditions, such as improved medical access, and enhanced community services [1]. Therefore, studying the relationships between social vulnerability and cognitive impairment could provide some guidance in reducing the adverse impacts of social vulnerability on elderly cognitive impairment in situations where it is difficult to change the social factors.

Previous research has examined the social factors that can contribute to cognitive decline, such as living situations [4,5], socioeconomic status [6,7], social support [8,9,10], social engagement and social networks [11,12]. For example, socio-demographic factors such as education and income [6] have been found to be beneficial in preventing cognitive decline, social engagement and social support have been found to be associated with a decreased risk of cognitive impairment [9,10], and married elderly [5,6] have been found to have a lower risk of cognitive impairment than those who are divorced, separated, or single. Although these studies supported the association between various social factors and cognitive impairment in the elderly, they generally neglected to examine the internal relationships between these social factors. Several studies that have considered the comprehensive social factors related to elderly health have used social vulnerability to describe the degree to which a person’s holistic social situation leaves them susceptible to health problems [8,13,14]. However, little cognitive impairment research has harnessed social vulnerability to examine the effects of multiple social factors [8]. Andrew et al. [8] used a logistic regression model to determine the associations between social vulnerability and cognitive decline and to measure the average effect of social vulnerability on cognitive decline. Focused on comprehensive social circumstances, this study also used social vulnerability as a contributory factor.

Previous studies have generally obtained the coefficients using ordinary least squares (OLS) in the linear regression models [3,4,5,6,7,15]. However, OLS can only provide a partial view of the relationships as the relationships at different points in the conditional distribution of the dependent variables [16] cannot be described, and the robustness of the OLS results can decrease when there are outliers and a non-normal distribution [16,17,18].

This study combined the holistic social factors to measure the social vulnerability, and employed an unconditional quantile regression model (UQR) on public survey data to investigate the relationships between social vulnerability and cognitive impairment. The relationships were also examined in different subsamples divided by different labels of a specific covariate, such as gender and age. Therefore, the main contributions of this paper are: (1) using social vulnerability as a contributory factor to describe holistic social factors; (2) applying an unconditional quantile regression to determine the non-linear relationships between social vulnerability and cognitive impairment; and (3) finding the elderly characteristics that affect the relationships between social vulnerability and cognitive impairment.

## 2. Methods

### 2.1. Study Sample

The data used in this study were taken from the 2014 Chinese Longitudinal Healthy Longevity Survey (CLHLS) [19], which was conducted to gain information on the health status and quality of life of the elderly including centenarians and nonagenarians in 22 Chinese provinces. The samples used in this study were all on people older than 60. The sample size of this survey was 6081, 48.6% of whom were male. For the elderly with cognitive impairment, their answers from self-reports can cause a methodological error. To deal with this problem, the interviewers firstly identified those questions that could not have valid answers through self-reports. Then, these questions were answered by other people. Overall, 63.6% of the elderly answered their own questions, while they others were answered mainly by their spouse (3.2%), child or spouse of child (25.6%), grandchild or spouse of grandchild (2.9%). In total, 95.62% of the proxy respondents live with the elderly and they were familiar with the elderly to the most detail. The survey gathered information on demographics and background (age, gender and occupation), health status (chronic diseases, functional limitations and cognitive impairments), socioeconomic characteristics (income, financial support and economic status), lifestyle (diet and nutrition, and drinking and smoking habits) and other information.

### 2.2. Measures

#### 2.2.1. Measure of Cognitive Impairment (Dependent Variable)

To measure the cognitive status, the questionnaire used in this survey used the Mini Mental State Examination (MMSE) [20], which had 30 scores in five categories: orientation, registration attention, calculation, recall, and language. This study calculated the cognitive impairment from the 0–30 scores, with the higher the score the worse cognitive function, to determine the factors with the positive coefficients in the regression results that aggravated elderly cognitive impairment, and the factors with the negative coefficients in the regression results that alleviated elderly cognitive impairment. As this study focused on people with mild, moderate or severe cognitive impairments, samples with MSSE scores lower than 5 were not included.

#### 2.2.2. Measure of Social Vulnerability (Independent Variable)

This study measured the social vulnerability by the sum of social vulnerability deficits summarized by Andrew et al. [8]. The deficits related to social factors were based on two principles [8,13]. First, the selected variables needed to include a broad representation of social factors. Second, the social vulnerability measures needed to be as sensible and as broadly applicable and comparable between the datasets as possible because of the constraints of the secondary data analysis. This study referred to selected variables from previous research and followed two basic principles. From the CLHLS questionnaire, 25 deficit items were included for the social vulnerability variable (Table 1). These deficit items were as binary values for the binary social deficits and as intermediate values for the ordered responses; higher scores indicated cumulated deficits and higher social vulnerability.

Social vulnerability is generally calculated as a proportion of the total number of deficit items by dividing the sum of deficit scores by the number of deficits considered (25 in this study). If the social vulnerability is scaled into [0,1], the coefficients of the social vulnerability will show a change from one extreme to the other. Compared with this extreme change in social vulnerability, it is more meaningful for us to study the changes of the cognitive impairment when the social vulnerability increases one deficit. Therefore, this study calculated the social vulnerability as the sum of the deficit scores and did not divide the scores by the number of deficits considered. Then, the social vulnerability ranged from 0 to 25.

#### 2.2.3. Measure of Covariates

Previous empirical studies identified several factors associated with cognitive impairment [6,7,8], which were included in the social vulnerability measure. Therefore, other factors were selected from the questionnaire as covariates: age and gender [6,8], smoking and drinking [6,21,22], exercise [23], depression [6], self-rated health [24], physical function [3,23], chronic disease [6], fruit and vegetable intake [25,26], vitamin intake [6], milk intake [6] and tea intake [27]. Gender was coded as 1 (male) and 2 (female), and depression was assessed using two variables (“feel sad, blue, or depressed” and “lost interest in most things that usually give you pleasure”), which were coded 1 for yes and 0 for no, with higher scores indicating a higher incidence of depression. The self-rated health was coded from 1 (very bad) to 5 (very good), and the physical function was assessed using five selected variables from the questionnaire, such as upper extremities, the ability to stand up without help, and the ability to use chopsticks, with higher scores indicating greater functional limitations. Chronic disease was measured by the number of chronic diseases the person had had or were currently experiencing. The vegetable intake and the fruit intake was scaled from 1 (rarely or never) to 4 (everyday or almost everyday). The vitamin, milk, and tea intakes were all included as covariates, with higher scores indicating a higher intake. As shown in Table 2, the elderly had a low average intake of vitamin, milk and tea. The smoking, drinking and exercise status were assessed by the accumulated years of smoking, drinking and exercise. In this survey, the data of four questions were used to calculate the accumulated years: (1) smoke at the present time (yes/no); (2) smoked in the past (yes/no); (3) the age when you began to smoke (denoted by *b*); and (4) the age when you stopped smoking if you do not smoke at the present time (denoted by *s*). The age at the time of completion of this survey was denoted by *a*. If the elderly smoke at the present time, their accumulated years of smoking were a−b. If the elderly smoked in the past and do not smoke at the present time, the accumulated years were s−b. If the elderly did not smoke in the past and also do not smoke at the present time, the accumulated years were 0. The calculation of the cumulative years of drinking and exercising were same as the calculation for smoking.

The description statistics about the dependent variable, independent variable and covariates were shown in Table 2.

## 3. Statistical Analysis

Section 2.2.3 reviews the factors related to cognitive impairment. This study first assumed the simple relationship that elderly *i*’s cognitive impairment (CI) was affected by 14 factors ($SV,X_{2}-X_{14}$): social vulnerability (SV), age, gender, lifestyle (smoke, drink, and exercise), health-related variables (self-rated health, depression, chronic disease, snd physical limitation), and diet (fruit and vegetable intake, vitamin intake, milk intake, and tea intake). The function of this assumption is shown in Equation (Equation 1), in which $\beta_{i}$ is the coefficients for the independent variables, and $\epsilon_{i}$ is the error term.

(1)$$CI_{i}=\beta_{0}+\beta_{1}SV_{i}+\sum_{i=2}^{13}\beta_{i}X_{i}+\epsilon_{i}$$

Then, the data were partitioned into different subsamples to measure different versions of Equation (Equation 1) in respect to gender, smoking habits, self-rated health level and other covariates. For example, the samples were partitioned into two groups by gender, and the function of the male subsample ($X_{2}=1$) was as shown in Equation (Equation 2).

(2)$$CI_{i}{|}_{X_{2}=1}=\beta_{0}^{X_{2}=1}+\beta_{1}^{X_{2}=1}SV_{i}{|}_{X_{2}=1}+\sum_{i=2}^{13}\beta_{i}^{X_{2}=1}X_{i}{{|}_{X_{2}=1}+\epsilon_{i}|}_{X_{2}=1}$$

This study used the regression models to explore the association between the social vulnerability and cognitive impairment. The coefficients of the independent variable, social vulnerability, can show how cognitive impairment changes when social vulnerability changes, but cannot show the changes of social vulnerability are the cause of changes of cognitive impairment. The causality relationship between social vulnerability and cognitive impairment can not be shown from the regression results, and is not discussed in this study. Our analysis only focused on the $\beta_{1}$ in Equations (Equation 1) and (Equation 2). This study measured the social vulnerability by the sum of the social vulnerability deficits, and $\beta_{1}$ showed the change of the cognitive impairment when the social vulnerability increased one deficit. In conclusion, we aimed to explore the association between social vulnerability and cognitive impairment under these conditions: different quantiles of cognitive impairment and different levels of a specific covariate (e.g., different gender and different age group).

The UQR estimator was used to measure the relationship between social vulnerability and cognitive impairment for the following reasons. First, compared with OLS, the quantile regression estimator does not require a normal distribution of the sample data [18,28,29,30], thus is more practical for empirical studies, and is also able to measure the different relationships at all the cognitive impairment quantiles. Second, compared to a conditional quantile regression model (CQR), the unconditional quantile regression model has been found to have better properties [18] as the CQR is only able to measure the relationship between a quantile of the dependent variable (e.g., the 5th quantile of cognitive impairment) and the independent variable (social vulnerability), conditional on the specific values of the other covariates [28]; however, the coefficients obtained using the UQR can be interpreted as the relationships between the social vulnerability and the cognitive impairment with all other covariates as constant. This means that the UQR measures the relationship between social vulnerability and cognitive impairment at the quantiles regardless of all the other covariates [16,17]; therefore, UQR results are more policy-focused as it measures the unchangeable relationships of the main variables at different conditional quantiles [16]. The Gaussian kernel density function was used to estimate the distribution of the cognitive impairment and bootstrapped standard errors, with all data manipulation and estimation being performed using Software *R*.

## 4. Results and Discussion

The results (Table 3) show that the coefficients of the full samples were all positive and statistically significant at all cognitive impairment quantiles. As shown in Figure 1, the relationship between social vulnerability and cognitive impairment became statistically stronger from the 50th to the 90th quantile of cognitive impairment; however, the coefficients from the 5th to 50th quantile were also statistically significant but very small. This pattern indicated that the positive relationship between social vulnerability and cognitive impairment was not constant and became larger as the cognitive impairment increased (50th–95th). Therefore, the more severe is the cognitive impairment of the elderly, the stronger is the relationship between social vulnerability and cognitive impairment exists.

The coefficients for the different subsamples are shown in Table 4, and the changes in the coefficients are shown in Figure 2. The bold type in Table 4 indicates the largest coefficient in the subsamples divided by the different levels of a specific covariate. For example, 0.057 in Table 4 was the largest coefficient in the different age groups. The coefficients of the social vulnerability were all positive in each subsamples, which indicated that the elderly who were more social vulnerable were more likely to have more severe cognitive impairment. According to the results for the different age subsamples, the coefficients for the elderly older than 88 was larger than the other elderly at all quantiles, which indicated that the cognitive impairment of these older people was more related to social vulnerability. For the subgroups divided by the different genders, the cognitive decline of the female elderly was found to be more related to social vulnerability than the male elderly. At the 95th quantile of cognitive impairment, the coefficient for the female was the largest. Previous research has identified female elderly at higher risk of cognitive impairment than male elderly. Therefore, these results add to the previous result: the female elderly are more likely to have cognitive impairment, and their cognitive impairment is more related to social vulnerability.

The subsamples divided by the health-related covariates also showed different strength of the positive relationships between social vulnerability and cognitive impairment. The coefficients indicated that, at all quantiles of cognitive impairment, the relationship became stronger in elderly who had lower self-rated health, higher depression levels, and more chronic diseases. For physical limitations, this relationship was found to be stronger for the people who had the worse physical functions (more physical limitations) up to 75th quantile. Previous research also found that that elderly who had lower self-rated health had higher risks of cognitive impairment [24]. Similarly, this result was also found in the elderly who were more depressed, had more chronic diseases, and had greater degree of physical limitations. Therefore, the findings in this study add to these previous results [6,23], as it was observed that these elderly (lower self-rated health, higher rates of depression, more chronic diseases, and more physical limitations) not only had a higher risk of cognitive impairment but also had cognitive impairment which were more related to social vulnerability.

Diet was found to affect the relationship between social vulnerability and cognitive impairment. Specifically, the cognitive impairment of the elderly who ate more fruit and vegetables was less related to social vulnerability at all quantiles. However, drinking more milk also weakened this relationship for elderly at the 90th and 95th quantile of cognitive impairment, and the other quantiles had small differences. Having more tea weakened this relationship from the 50th to 95th quantile, and the other quantiles had small differences. As the subsamples for higher or lower vitamin intake showed only small differences in the relationship, the result was not included in the figure. Fruit and vegetables intake were found to affect the relationship between social vulnerability and cognitive impairment at all impairment levels; however, milk and tea consumption affected this relationship significantly only in the elderly with more severe cognitive impairments. Previous research has found that the consumption of fruit and vegetables [25,26], milk [6], and tea [27] reduced the risk of cognitive impairment. Therefore, the results in this study add to these results, as it was found that a good diet not only reduced the risk of cognitive impairment, but also reduced the relationship between the elderly’s cognitive impairment and social vulnerability.

The lifestyle factors were found to have varying effects on the relationship between cognitive impairment and social vulnerability. Elderly exercisers were found to have less cognitive impairment which was less related to social vulnerability from the 5th quantile to the 90th quantile. Previous research also found that exercise was negatively related to cognitive decline. Therefore, the results in this study add to the previous results, as it was found that not only could exercise decrease the risk of cognitive impairment [23], but it could also reduce the relationship between the elderly’s cognitive impairment and the social vulnerability.

The positive relationship between the social vulnerability and the cognitive impairment was much stronger for the nonsmokers and nondrinkers than the smokers and drinkers. Previous research has found that smoking and drinking increased the risk of cognitive impairment [21,22]. This means that, although nonsmokers and nondrinkers have a lower risk of cognitive impairment, they have a larger relationship between their cognitive impairment and social vulnerability. To interpret this result, the characteristics of the smokers and drinkers were further investigated. For the nonsmokers, their average cognitive impairment score was 4, which was higher than that of smokers, which was 3. According to Figure 1, the strength of the relationship between the social vulnerability and the cognitive impairment was positively related to the cognitive impairment. Therefore, for the nonsmokers and nondrinkers, the relationship between social vulnerability and cognitive impairment was much stronger than that of smokers and drinkers.

In conclusion, the results show that there was an overall non-linear pattern in the positive relationships between social vulnerability and cognitive impairment. The relationships were stronger at higher levels of the cognitive impairment distribution, and more significant for those elderly with the following characteristics: older age, female, lack of exercise, lower self-rated health, higher depression, more physical limitations, more chronic diseases, and consume less fruit, vegetables, milk and tea. Therefore, targeting efforts toward the elderly whose cognitive impairments are the most related to social vulnerability would have the highest effectiveness. The elderly can reduce the relationship between their social vulnerability and cognitive decline by doing more exercise, alleviating their depression, eating more fruit and vegetables, and drinking more milk and tea.

## 5. Robustness Checks

This study used the approach in [31] for the robustness checks. The robustness check used different kernel density functions and compared the differences in the coefficients at the same quantiles. The coefficients were originally estimated using the Gaussian (Gau.) kernel; therefore, the Optcosine (Optcos.) and Biweight (BW) kernels were chosen for the robustness check. The estimated coefficients obtained at the 10th, 50th and 90th quantiles are shown in Table 5. As the results using the different kernels did not have any significant differences, the comparison provided support that the estimates in this study did not have spurious relationships.

## 6. Conclusions

This study explored the relationship between social vulnerability and cognitive impairment using an unconditional quantile regression model. The non-linear positive relationship indicates that the strength of this relationship was different for the elderly who had different degrees of cognitive impairment; that is, the relationship between cognitive impairment and social vulnerability were found to be stronger for the elderly who had more severe cognitive impairments. This study also examined the relationship of different subsamples focused on lifestyle, diet and other health characteristics, from which it was found that cognitive impairment more strongly related to social vulnerability when the elderly were female; had less exercise; had lower self-rated health; had greater incidences of depression, chronic disease and physical limitations; and consumed less fruit, vegetable, milk and tea. These results could be used to provide guidance to the elderly to decrease their risk of cognitive impairment. They can also give some insights for the experts to conduct experiments about the cognitive impairment of the elderly.

Despite the contributions of this paper, it also has some limitations. First, the MMSE used to measure the cognitive impairment did not include the assessment of executive function, which can influence the relationships between social vulnerability and cognitive impairment. For example, the executive dysfunction of the elderly may be specifically related to social vulnerability through the frontal lobe function. Second, for the elderly with cognitive impairments, some questions were answered by proxy respondents, which may cause some deviations from reality. Third, the survey was designed and executed in China. The results in different regions may be different, and it is meaningful that the researchers in other regions also use the unconditional quantile regression model to find some differences.

## Figures and Tables

**Figure 1 ijerph-16-03684-f001:**
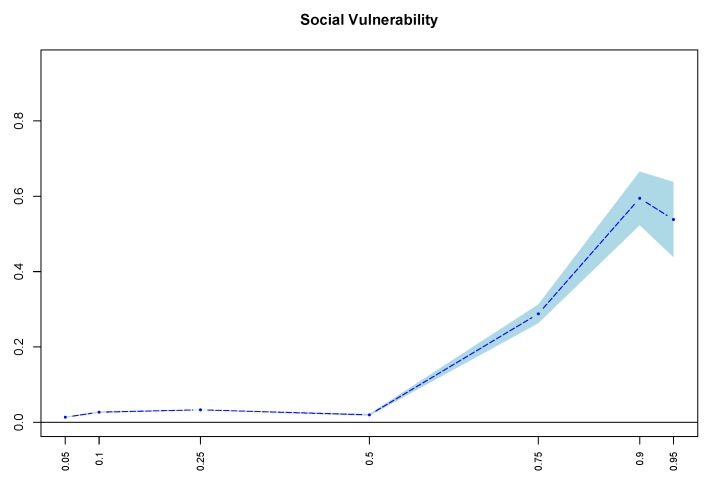
Coefficients of social vulnerability in full samples. Note: The horizontal axis represents quantiles of cognitive impairment, while the vertical axis represents coefficients of the social vulnerability variable. Bootstrapped standard error bars using 100 repetitions are reported. with blue shading

**Figure 2 ijerph-16-03684-f002:**
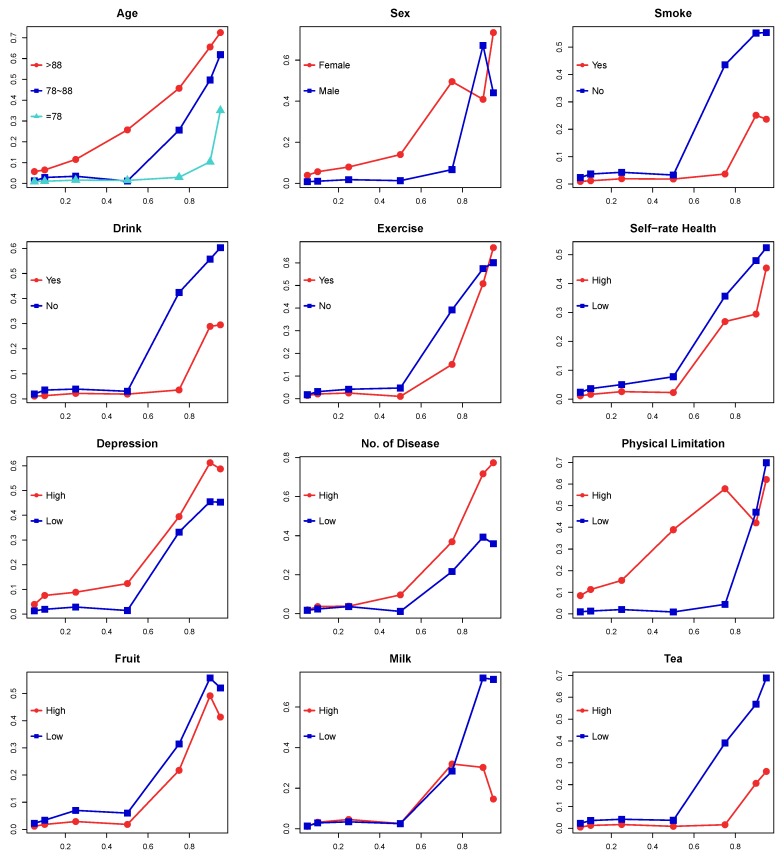
Coefficients of social vulnerability in subsamples. Note: Horizontal axis: quantiles of cognitive impairment, Vertical axis: coefficients of the social vulnerability variable.

**Table 1 ijerph-16-03684-t001:** Items of the Social Vulnerability.

ITEM (SCORE)	MEAN VALUE
Living situation:	
**1. Current marital status**	0.5691
currently married and living with spouse (0)	
married but not living with spouse (0.5)	
divorced/widowed/never married(1)	
**2. Co-residence**	0.2204
with household member(s) (0)/alone (1)	
Social support:	
**3. Can get adequate medical service when you are sick**	0.0323
yes (0)/no (1)	
**4. Social service level in your community (5 levels)**	0.8013
very good (0)/good (0.25)/so so (0.5)/bad (0.75)/very bad(1)	
**5. Residence**	0.7019
city (0)/town (0.5)/rural (1)	
**6. Feel need more home visits**	0.8272
yes (1)/no (0)	
**7. Feel need more social and recreation activities**	0.6433
yes (1)/no (0)	
**8. Feel need more neighboring relations**	0.6419
yes (1)/no (0)	
Socially oriented Activities of Daily Living:	
**9. Visit your neighbors by yourself**	0.1384
yes, independently (0)/yes, but need some help (0.5)/no, can’t (1)	
**10. Go shopping by yourself**	0.2266
yes, independently (0)/yes, but need some help (0.5)/no, can’t (1)	
**11. Walk continuously for 1 kilometer at a time by yourself**	0.3163
yes, independently (0)/yes, but need some help (0.5)/no, can’t (1)	
**12. Take public transportation by yourself**	0.3723
yes, independently (0)/yes, but need some help (0.5)/no, can’t (1)	
Social engagement and leisure:	
**13. Tours beyond home city/county have you made in the past two years**	0.9913
Normalize data to 0–1 number	
**14. Garden work**	0.8055
almost everyday (0)/once for a week (0.25)/at least once for a month (0.5)	
/not monthly, but sometimes (0.75)/never (1)	
**15. Play cards or mah-jong**	0.8635
almost everyday (0)/once for a week (0.25)/at least once for a month (0.5)	
/not monthly, but sometimes (0.75)/never (1)	
**16. Social activities (organized)**	0.9177
almost everyday (0)/once for a week (0.25)/at least once for a month (0.5)	
/not monthly, but sometimes (0.75)/never (1)	
Empowerment, life control:	
**17. Make your own decisions concerning your personal affairs**	0.3064
always (0)/often (0.25)/sometimes (0.5)/seldom (0.75)/never (1)	
**18. Feel that you are a person of worth at least equal to others**	0.4443
always (0)/often (0.25)/sometimes (0.5)/seldom (0.75)/never (1)	
**19. Take a positive attitude towards yourself**	0.2835
always (0)/often (0.25)/sometimes (0.5)/seldom (0.75)/never (1)	
**20. Status of decision making on financial spending in your household**	0.4923
almost all spending in my household (0)	
some of the main spending in my household (0.25)	
some of the non-main spending in my household (0.5)	
only on own spending (0.75)	
cannot make decisions on any spending (1)	
Socio-economic status:	
**21. Total income of your household last year continuous**	0.6206
Normalize data to 0–1 number	
**22. All of the financial support is sufficient to pay for daily expenses**	0.1712
yes (0)/no (1)	
**23. Rate your economic status compared with other local people**	0.4844
very rich (0)/rich (0.25)/so so (0.5)/poor (0.75)/very poor (1)	
**24. Educational attainment**	0.8708
Normalize data to 0-1 number	
**25. Occupation**	0.9121
white-collar (0)/non-white-collar (1)	

**Table 2 ijerph-16-03684-t002:** Descriptive Statistics (N=6081).

	Mean	Variance	Min	Max
social vulnerability	13.6549	9.4382	3	23
age	83.8831	100.7362	60	114
self-health rate	3.3894	0.7573	1	5
smoke	15.7709	640.1702	0	99
drink	12.6304	548.3681	0	101
exercise	9.4722	345.5731	0	99
depression	0.2003	0.2599	0	2
physical limitation	0.3265	0.5312	0	5
chronic disease	0.6227	0.8340	0	10
vegetable and fruit	2.8740	0.4720	1	4
vitamin intake	1.4769	1.2019	1	5
milk intake	2.5549	2.4272	1	5
tea intake	2.1793	2.8196	1	5
cognitive impairment	10.3247	21.9063	5	30

**Table 3 ijerph-16-03684-t003:** Unconditional quantile regression and OLS results

*Cognitive Impairment* ${}^{d.v.}$	5th	10th	25th	50th	75th	90th	95th	OLS
$social\;vulnerability{\;}^{i.d.v.}$	0.0135 ***	0.0270 ***	0.0331 ***	0.0197 ***	0.2880 ***	0.5945 ***	0.5383 ***	0.1727 ***
$age{\;}^{c.v.}$	0.0025 ***	0.0054 ***	0.0079 ***	0.0242 ***	0.1605 ***	0.2459 ***	0.2633 ***	0.0948 ***
$gender{\;}^{c.v.}$	0.0105 ***	0.0110 ***	0.0390 ***	0.2446 ***	1.4684 ***	2.0544 ***	1.0007 ***	0.7869 ***
$smoke{\;}^{c.v.}$	0.0003 ***	0.0002 ***	0.0003 ***	0.0010 ***	0.0009	−0.0137 ***	−0.0179 ***	−0.0023
$drink{\;}^{c.v.}$	0.0002 ***	0.0004 ***	0.0004 ***	0.0005 ***	0.0031 ***	0.0016 ***	−0.0083	0.0007
$exercise{\;}^{c.v.}$	0.0004 ***	−0.0004 ***	−0.0005 ***	−0.0010 ***	−0.0131 ***	−0.0051 ***	−0.0182 ***	−0.0055
$self-rated\;health{\;}^{c.v.}$	−0.0079 ***	−0.0273 ***	−0.0121 ***	−0.0731 ***	−0.3022 ***	−0.3023 ***	−0.0077	−0.1171
$depression{\;}^{c.v.}$	0.0090 ***	0.0200 ***	0.0360 ***	0.1432 ***	0.3713 ***	0.5915 ***	1.3295 ***	0.3987 ***
$chronic\;disease{\;}^{c.v.}$	0.0064 ***	0.0025 **	0.0076 ***	0.0490 ***	0.3226 ***	0.0559 *	0.0400	0.1170 *
$physical\;limitation{\;}^{c.v.}$	−0.0084 ***	−0.0027 *	0.0150 ***	0.1268 ***	1.3693 ***	3.0656 ***	4.0602 ***	1.0983 ***
$vegetable\;and\;fruit{\;}^{c.v.}$	−0.0046 ***	−0.0166 ***	−0.0210 ***	−0.0789 ***	−0.3635 ***	−0.8493 ***	−0.9600 ***	−0.2778 ***
$vitamin\;intake{\;}^{c.v.}$	−0.0018	0.0155 ***	0.0145 ***	0.0218 ***	0.0558 *	−0.1929 ***	−0.2034 ***	−0.0148
$milk\;intake{\;}^{c.v.}$	−0.0004	−0.0073 ***	−0.0053 ***	0.0114 ***	0.0477 ***	0.0736 ***	0.1947 ***	0.0494
$tea\;intake{\;}^{c.v.}$	−0.0161	−0.0273	−0.0284	−0.0154	−0.1035	0.0093	−0.1728	−0.0412
intercept	0.6825 ***	0.4585 ***	0.3545 ***	−0.8739 ***	−12.0662 ***	−17.4808 ***	−13.7165 ***	−6.2805 ***

* Parameters statistically different from 0 at the 10% significant level. ** Parameters statistically different from 0 at the 5% significant level. *** Parameters statistically different from 0 at the 1% significant level. d.v. Dependent variable. i.d.v. Independent variable. c.v. Covariates.

**Table 4 ijerph-16-03684-t004:** Estimated relationship of social vulnerability on cognitive impairment across subsamples

Sample	5th	10th	25th	50th	75th	90th	95th
Full sample	0.014 ***	0.027 ***	0.033 ***	0.020 ***	0.288 ***	0.594 ***	0.538 ***
age > 88	0.057 ***	0.065 ***	0.115 ***	0.257 ***	0.457 ***	0.656 ***	0.725 ***
78 < age <= 88	0.012 ***	0.029 ***	0.034 ***	0.011 ***	0.256 ***	0.497 ***	0.619 ***
age <= 78	0.008 ***	0.010 ***	0.016 ***	0.015 ***	0.029 ***	0.104	0.351
female	0.040 ***	0.057 ***	0.080 ***	0.140 ***	0.495 ***	0.409 ***	0.734 ***
male	0.008 ***	0.011 ***	0.018 ***	0.013 ***	0.067 ***	0.671 ***	0.441 ***
no smoking	0.023 ***	0.036 ***	0.043 ***	0.033 ***	0.435 ***	0.551 ***	0.553 ***
smoking	0.009 ***	0.012 ***	0.019 ***	0.018 ***	0.036 **	0.251 ***	0.237 **
no drinking	0.019 ***	0.035 ***	0.039 ***	0.030 ***	0.424 ***	0.557 ***	0.603 ***
drinking	0.011 ***	0.013 ***	0.022 ***	0.019 ***	0.035 **	0.289 ***	0.295 ***
no exercising	0.017 ***	0.031 ***	0.041 ***	0.047 ***	0.392 ***	0.575 ***	0.601 ***
exercising	0.014 ***	0.021 ***	0.025 ***	0.010 ***	0.151 ***	0.508 ***	0.667 ***
low fruit and vegetable intake	0.022 ***	0.034 ***	0.070 ***	0.060 ***	0.314 ***	0.557 ***	0.520 ***
high fruit and vegetable intake	0.012 ***	0.018 ***	0.029 ***	0.018 ***	0.217 ***	0.492 ***	0.413 ***
low self-rate health	0.024 ***	0.037 ***	0.050 ***	0.078 ***	0.356 ***	0.479 ***	0.524 ***
high self-rate health	0.012 ***	0.016 ***	0.026 ***	0.023 ***	0.268 ***	0.294 ***	0.454 ***
low depression	0.013 ***	0.020 ***	0.029 ***	0.015 ***	0.332 ***	0.454 ***	0.453 ***
high depression	0.039 ***	0.076 ***	0.089 ***	0.124 ***	0.394 ***	0.612 ***	0.587 ***
less phsical function limitation	0.009 ***	0.013 ***	0.020 ***	0.009 ***	0.044 ***	0.470 ***	0.699 ***
more physical functional limitations	0.085 ***	0.113 ***	0.155 ***	0.389 ***	0.579 ***	0.420 ***	0.621 ***
less chronic diseases	0.017 ***	0.024 ***	0.036 ***	0.012 ***	0.216 ***	0.392 ***	0.359 ***
more chronic diseases	0.020 ***	0.037 ***	0.038 ***	0.096 ***	0.369 ***	0.716 ***	0.773 ***
low vitamin intake	0.013 ***	0.031 ***	0.033 ***	0.024 ***	0.312 ***	0.521 ***	0.656 ***
high vitamin intake	0.020 ***	0.025 ***	0.043 ***	0.016 ***	0.365 ***	0.515 ***	0.419 **
low milk intake	0.013 ***	0.029 ***	0.034 ***	0.025 ***	0.284 ***	0.743 ***	0.736 ***
high milk intake	0.012 ***	0.032 ***	0.046 ***	0.026 ***	0.319 ***	0.303 ***	0.147
low tea intake	0.023 ***	0.036 ***	0.042 ***	0.037 ***	0.391 ***	0.569 ***	0.688 ***
high tea intake	0.006 ***	0.013 ***	0.018 ***	0.009 **	0.017	0.206 ***	0.261 **

* Parameters statistically different from 0 at the 10% significant level. ** Parameters statistically different from 0 at the 5% significant level. *** Parameters statistically different from 0 at the 1% significant level.

**Table 5 ijerph-16-03684-t005:** Robustness checks: UQR results using different kernel density functions.

	10th			50th			90th		
	**Gau.**	**Optcos.**	**BW**	**Gau.**	**Optcos.**	**BW**	**Gau.**	**Optcos.**	**BW**
Social vulnerability	0.03 ***	0.03 ***	0.03 ***	0.02 ***	0.02 ***	0.02 ***	0.59 ***	0.51 ***	0.53 ***
Age	0.01 ***	0.01 ***	0.01 ***	0.02 ***	0.03 ***	0.03 ***	0.25 ***	0.21 ***	0.22 ***
Gender	0.01 ***	0.01 ***	0.01 ***	0.24 ***	0.28 ***	0.27 ***	2.05 ***	1.76 ***	1.84 ***
Smoke	0.00 ***	0.00 ***	0.00 ***	0.00 ***	0.00 ***	0.00 ***	−0.01 ***	−0.01 ***	−0.01 ***
Drink	0.00 ***	0.00 ***	0.00 ***	0.00 ***	0.00 ***	0.00 ***	0.00 ***	0.00 ***	0.00 ***
Exercise	0.00 ***	0.00 ***	0.00 ***	0.00 ***	0.00 ***	0.00 ***	−0.01 ***	0.00 ***	0.00 ***
Self-rate health	−0.03 ***	−0.03 ***	−0.03 ***	−0.07 ***	−0.08 ***	−0.08 ***	−0.30	−0.26	−0.27
Depression	0.02 ***	0.02 ***	0.02 ***	0.14 ***	0.16 ***	0.16 ***	0.59 ***	0.51 ***	0.53 ***
Chronic disease	0.00 ***	0.00 ***	0.00 ***	0.05 ***	0.06 ***	0.05 ***	0.06 ***	0.05 ***	0.05 ***
Physical limitation	0.00 ***	0.00 ***	0.00 ***	0.13 ***	0.14 ***	0.14 ***	3.07 ***	2.63 ***	2.74 ***
Fruit and	−0.02 **	−0.02 **	−0.02 **	−0.08 ***	−0.09 ***	−0.09 ***	−0.85 *	−0.73	−0.76
vegetable intake
Vitamin intake	0.02 ***	0.02 ***	0.02 ***	0.02 ***	0.02 ***	0.02 ***	−0.19 ***	−0.17 ***	−0.17 ***
Milk intake	−0.01 ***	−0.01 ***	−0.01 ***	0.01 ***	0.01 ***	0.01 ***	0.07 ***	0.06 ***	0.07 ***
Tea intake	−0.03 ***	−0.03 ***	−0.03 ***	−0.02 ***	−0.02 ***	−0.02 ***	0.01 ***	0.01 ***	0.01 ***
Intercept	0.46	0.37	0.39	−0.87	−1.17	−1.10	−17.48	−13.64	−14.63

* Parameters statistically different from 0 at the 10% significant level. ** Parameters statistically different from 0 at the 5% significant level. *** Parameters statistically different from 0 at the 1% significant level.

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
