# Peer review of "The Relationship between Cognitive Impairment and Social Vulnerability among the Elderly: Evidence from an Unconditional Quantile Regression Analysis in China"

_ijerph, 2019, doi:10.3390/ijerph16193684_

Round 1
Reviewer 1 Report
This is an interesting analysis the relation between social vulnerability and cognitive impairment. But it makes me confusing in
1.authors say coefficient is the effect of unit change of social vulnerability on cog impairment. but the results are more cognitive imparied, more social vulnerable. With a similar reason, please describe in detail about independent factor and depdendent factor on the figure and table legend.
2. this is cross-sectional study, but authors insist high Cognitive impairment is susceptible to or do affect on social vulnerability. How can you insist on the relation cause and effect?
Author Response
Response to Reviewer 1
We have read the comments from you very carefully. Accordingly we have made a studious attempt to revise the paper thoroughly. We are grateful to you for your suggestions in improving this paper. Certainly, it has helped us to clarify several issues and hence, improved the paper.
We then give point-to-point response to the comments in the following, where the comments are marked with *. In the paper, we used red color to highlight the revised parts.
* This is an interesting analysis the relation between social vulnerability and cognitive impairment. But it makes me confusing in:
*authors say coefficient is the effect of unit change of social vulnerability on cog impairment. but the results are more cognitive imparied, more social vulnerable. With a similar reason, please describe in detail about independent factor and depdendent factor on the figure and table legend.
[Our Response]: We are sorry to make you confused about that. This study focused on the relationship between social vulnerability and cognitive impairment. The results we obtained is that the relationship between social vulnerability and cognitive impairment is larger for the more cognitive impaired elder. To make it clear, we have added the description of the independent and dependent factors on the section titles: “Measure of cognitive impairment (dependent variable)'' and “Measure of social vulnerability (independent variable)''. Besides, we also added the description of independent and dependent variables on the legend of Table 2.
* this is cross-sectional study, but authors insist high Cognitive impairment is susceptible to or do affect on social vulnerability. How can you insist on the relation cause and effect?
[Our Response]: Thanks a lot for your comment. We are so sorry to make you confused about that. This study explored the correlation but not the causal relationship. We revised the wrong descriptions in the original version, and added one more paragraph to explain it: ``This study used the regression models to explore the correlation between the social vulnerability and cognitive impairment. The coefficients of the independent variable, social vulnerability, can show how cognitive impairment changes when social vulnerability changes, but can not show the changes of social vulnerability are the cause of changes of cognitive impairment. Our analysis focused on the \beta_1 in Equation (1) and Equation (2), which showed the change of the cognitive impairment when the social vulnerability changes one unit. In conclusion, we aimed to explore the correlation relationship between social vulnerability and cognitive impairment under these conditions: different quantiles of cognitive impairment and different levels of a specific covariate (e.g. different gender and different age group).'' On the conclusion section, we also revised the wrong descriptions as: “The non-linear positive relationship indicated that the strength of this relationship was different for the elderly who had different degrees of cognitive impairment; that is, the relationship between cognitive impairment and social vulnerability were found to be stronger for the elderly who had more severe cognitive impairments. This paper also examined the relationship of different subsamples focused on lifestyle, diet and other health characteristics, from which it was found that cognitive impairment more strongly related to social vulnerability when the elderly were female, had less exercise, had lower self-rated health, had greater incidences of depression, chronic disease and physical limitations and consumed less fruit, vegetable, milk and tea.''
Although we have made a very thorough revision to the paper, there may be still some problems. We sincerely appreciate your further recommendations to enhance the quality of this paper.
Thank you very much and best regards.
Reviewer 2 Report
Dear Authors,
thank You very much for this manuscript. It is well written, though it deals with a topic that is already widely evaluated in literature.
Here are my comments and suggestions :
> the Social Vulnerability Index (SVI) proposed by Andrew et al. is based on self-reports. In elderly patients with cognitive impairment, this can realize a methodological error. This point needs to be more detailed ;
> the relationship between SVI and severity of cognitive impairment can be bi-directional : one can influence the other, and vice versa. It is not clear how this point has been addressed in Your manuscript ;
> the fact that smoking and drinking did not take large effect on the cognitive impairment, leaves some legitimate doubts about the correctness of the methodology You used. Please, review Your data by combining ALL variables.
Best regards.
Author Response
Response to Reviewer 2
We have read the comments from you very carefully. Accordingly we have made a studious attempt to revise the paper thoroughly. We are grateful to you for your suggestions in improving this paper. Certainly, it has helped us to clarify several issues and hence, improved the paper.
We then give point-to-point response to the comments in the following, where the comments are marked with *. In the paper, we used blue color to highlight the revised parts.
* It is well written, though it deals with a topic that is already widely evaluated in literature.
* The Social Vulnerability Index (SVI) proposed by Andrew et al. is based on self-reports. In elderly patients with cognitive impairment, this can realize a methodological error. This point needs to be more detailed.
[Our Response]: Thanks a lot for your recommendation! This survey has considered the methodological error, and we are sorry not to explain that in the first manuscript. We have revised this paper to explain this point. We have added more information in Section 2.1.: “For the elderly who with cognitive impairment, their answers from self-reports can cause the methodological error. To deal with this problem, the interviewers identified those questions that could not obtain valid answers through self-reports. These questions were answered by other people. 63.6% of the elderly answered their own questions, and others' answers were mainly from their spouse (3.2%), child or spouse of child (25.6%), grandchild or spouse of grandchild (2.9%).''
* The relationship between SVI and severity of cognitive impairment can be bi-directional : one can influence the other, and vice versa. It is not clear how this point has been addressed in Your manuscript.
[Our Response]: We are sorry to make you confused about this. In this study, we only discussed the correlation relationship between social vulnerability and cognitive impairment. The causal relationship between them was not discussed in this paper. Specifically, we added a paragraph as follows: ``This study used the regression models to explore the correlation between the social vulnerability and cognitive impairment. The coefficients of the independent variable, social vulnerability, can show how cognitive impairment changes when social vulnerability changes, but can not show the changes of social vulnerability are the cause of changes of cognitive impairment. Our analysis focused on the \beta_1 in Equation (1) and Equation (2), which showed the change of the cognitive impairment when the social vulnerability changes one unit. In conclusion, we aimed to explore the correlation relationship between social vulnerability and cognitive impairment under these conditions: different quantiles of cognitive impairment and different levels of a specific covariate (e.g. different gender and different age group).''
*The fact that smoking and drinking did not take large effect on the cognitive impairment, leaves some legitimate doubts about the correctness of the methodology You used. Please, review Your data by combining ALL variables.
[Our Response]: Thanks a lot for you to find this problem. We reviewed our data and there was no mistake of the data. We used the years of the smoking and drinking, so the coefficients of smoke and drink were small values. The coefficients between smoking and drinking and cognitive impairment were significant at the 99% confidence interval. The wrong descriptions have been deleted, and in the section of the measure of covariates, we also described the smoking and drinking data to make it clear: “the smoking, drinking status were assessed by the accumulated years of smoking and drinking''.
Although we have made a very thorough revision to the paper, there may be still some problems. We sincerely appreciate your further recommendations to enhance the quality of this paper.
Thank you very much and best regards.
Round 2
Reviewer 2 Report
Dear Authors,
thanks for the revised version of Your manuscript.
All my comments and suggestions were satisfactorily met.
Quality of presentation has been significantly improved.
Minor point :
line 63 : from self-reports can cause a methodological error (and not : the methodological error).
With kind regards.
Author Response
Response
2019.09.17
We are very grateful to you for your suggestions in improving this paper! We then give point-to-point response to the comments in the following, where the comments are marked with *.
*line 63 : from self-reports can cause a methodological error (and not : the methodological error).
Our response: Thanks a lot for finding this mistake. We have replaced “the” with “a”: “For the elderly with cognitive impairment, their answers from self-reports can cause a methodological error.”
Thank you very much and best regards.